# Extracellular Vesicles in the Crosstalk of Autophagy and Apoptosis: A Role for Lipid Rafts

**DOI:** 10.3390/cells14100749

**Published:** 2025-05-20

**Authors:** Agostina Longo, Valeria Manganelli, Roberta Misasi, Gloria Riitano, Tuba Rana Caglar, Elena Fasciolo, Serena Recalchi, Maurizio Sorice, Tina Garofalo

**Affiliations:** Department of Experimental Medicine, “Sapienza” University of Rome, 00161 Rome, Italy; agostina.longo@uniroma1.it (A.L.); valeria.manganelli@uniroma1.it (V.M.); roberta.misasi@uniroma1.it (R.M.); gloria.riitano@uniroma1.it (G.R.); tubarana.caglar@uniroma1.it (T.R.C.); elena.fasciolo@uniroma1.it (E.F.); serena.recalchi@uniroma1.it (S.R.); maurizio.sorice@uniroma1.it (M.S.)

**Keywords:** extracellular vesicles, exosomes, apoptosis, autophagy, lipid rafts

## Abstract

Autophagy and apoptosis are two essential mechanisms regulating cell fate. Although distinct, their signaling pathways are closely interconnected through various crosstalk mechanisms. Lipid rafts are described to act as both physical and functional platforms during the early stages of autophagic and apoptotic processes. Only recently has a role for lipid raft-associated molecules in regulating EV biogenesis and release begun to emerge. In particular, lipids of EV membranes are essential components in conferring stability to these vesicles in different extracellular environments and/or to facilitate binding or uptake into recipient cells. In this review we highlight these aspects, focusing on the role of lipid molecules during apoptosis and secretory autophagy pathways. We describe the molecular machinery that connects autophagy and apoptosis with vesicular trafficking and lipid metabolism during the release of EVs, and how their alterations contribute to the development of various diseases, including autoimmune disorders and cancer. Overall, these findings emphasize the complexity of autophagy/apoptosis crosstalk and its key role in cellular dynamics, supporting the role of lipid rafts as new therapeutic targets.

## 1. Introduction

Extracellular vesicles (EVs) are a heterogeneous group of vesicles delimitated by a lipid bilayer that are released by all human cell types under both physiological and pathological conditions, for the purpose of regulating intercellular signaling networks through the transfer of active biomolecules, such as proteins, lipids, RNAs, DNA and microRNAs [1,2,3,4]. EVs are also produced “in vitro” under various cell culture conditions, thus creating cell culture-conditioned media [5].

Over the years, an increasing number of EV-related studies have been conducted, not only to understand their biogenesis and composition, but also to elucidate the molecular machinery underlying the specific release for each EV subtype. Despite this positive trend, overall research on EVs has often led to evidence of contradictory findings, probably due to differences in vesicles purification protocols, [6,7,8,9], differences in vesicles characterization techniques [10,11], and differences in the biological fluids [12,13] or cell types investigated [14,15,16], including cells cultured in vitro [17,18]. Understanding how different EV subtypes are generated has been a long-standing goal, as it is critical to discriminate them and to explore their diagnostic and therapeutic relevance. Anyhow, it is essential to consider that their functions and potential applications remain yet to be elucidated, and in this regard, identification of specific markers for the characterization and isolation of EV populations has a relevant role.

Though EVs have been characterized mainly in terms of their protein and nucleic acid content [19,20], lipidomic studies have only partially contributed to defining the prevalence of lipid species, as well as their important structural and regulatory functions during EVs’ biogenesis, release, targeting and cellular uptake [21,22,23,24].

In this review, we outlined the role of the interconnected pathways between autophagy and apoptosis as mandatory of extracellular vesicles, focusing on recent advances on the role of lipid rafts.

## 2. Classification of EVs

There is still no real consensus regarding EV classification and nomenclature [3], probably due to the variety in EVs’ size, composition, origin and targets, as well as to difficulties related to their isolation and analysis. However, the International Society of Extracellular Vesicles (ISEV) classifies the main EV subpopulations into exosomes, microvesicles and apoptotic vesicles [25].

Exosomes, which are approximately 50–100 nm in diameter, are the smallest type of EVs, and they originate inside the cell, within the endocytic pathway regulated mainly by the ESCRT complex, although an ESCRT-independent pathway has also been described [26]; moreover, they are generated by exocytosis (i.e., budding) of multivesicular bodies (MVBs) containing intraluminal vesicles (ILVs).

EVs that are larger than exosomes, ranging from approximately 100–1000 nm in diameter, are variably called “microvesicles” (MVs) or “microparticles” (MPs). They are generated by the outward budding process of the plasma membrane, which involves the physical alteration of the phospholipid membrane, strictly regulated by a sequence of calcium-dependent enzymatic steps, including scramblases, flippases and aminophospholipid translocase [27].

The third type of EV is referred to as apoptotic vesicles, which are generated by plasma membrane blebbing of cells undergoing apoptosis. In fact, whereas exosomes and MVs are secreted during normal cellular processes, apoptotic bodies (ApoBDs) are formed only during programmed cell death. In addition to traditional ApoBDs, smaller vesicles have been identified as apoptotic cell-derived extracellular vesicles (ApoEVs) [28] or exosome-like ApoEVs [29], produced during the breakdown of dying cells. Although there are no well-defined criteria to distinguish ApoBDs from other ApoEVs, vesicles can be classified by diameter: larger membrane-wrapped vesicles, termed ApoBDs, have diameters of 1000–5000 nm [30], while the smaller vesicles ApoEVs have diameters of 50–1000 nm [31,32].

It is important to consider that vesicles generated by different biogenic pathways have overlapping size ranges, suggesting that neither size nor the mechanism of biogenesis can be used to classify them in a satisfactory manner. Therefore, identification and validation by using specific markers could be helpful for more accurate classification. To date, some proposed vesicular markers include DNA or molecular components of cytosolic organelles for apoptotic vesicles, tetraspanins (i.e., CD63 and CD9) for exosomes, or integrin β1 for MVs [33].

Although EV classification has been mainly based on their size, origin and protein content, the protein/lipid ratio has also been proposed by some authors as an alternative criterion. For instance, apoptotic bodies exhibit the highest protein/lipid ratio, followed by MVs, and then by exosomes, as demonstrated by the characterization of EVs released from different myeloid or lymphoid cell lines, as well as blood plasma [34].

## 3. Functions of EVs

Many cell-intrinsic functions have been attributed to EVs that are only now beginning to be understood. Evidence indicates that EVs are crucial in cell–cell communications by shuttling complex messages in the form of membrane proteins, carbohydrates, lipids and other cargo molecules, including RNAs, proteins and metabolites, that require protection from extracellular enzymatic degradation. EVs also regulate cell signaling during the maturation of reticulocytes by the removal of plasma membrane receptors [35], the release of signaling molecules [36] and the transfer of microRNA (miRNA) [37]. Therefore, EVs are associated with a variety of cell–cell communication mediators, including cytokines [38,39], hormones and neurotransmitters [40]. As a matter of fact, EVs have been shown to contribute to a large variety of pathophysiological processes, including red blood cell senescence [41], inflammation [42,43], migration [44], immune system activation, suppression [45] and tumorigenesis [46]. In many cases, EV biogenesis and secretion have been linked to the activation of specific signaling pathways that include apoptosis or autophagy (Figure 1).

In particular, these vesicles serve as carriers of selected proteins, supporting both their biogenesis and release by carrying specific components, including some lipidic species, as will be explored further in this review.

## 4. Characterization of EVs

Recent advances in lipidomic studies have provided new information on the origin and characterization of EVs, confirming the presence of specific subsets of lipid classes somehow related to cellular variability and biological fluids, including plasma and urine [48]. Moreover, it is still not clear how the lipid profile of EVs is regulated, nor how the sphingolipid composition is regulated in the endosome pathway, and in EVs, this is yet to be investigated. The lipid composition of EVs differs from the lipid profile of the donor cells [22,49], but it may depend on the physiological stage of the producer cell and the fate and function of EVs [50,51]. Nevertheless, lipidomic analyses have revealed that the bilayer membrane of EVs can often be enriched in lipid components, such as phospholipids (including phosphatidylcholine, phosphatidylserine), and mainly (up to 3 times) ceramide, cholesterol, GSLs, sphingomyelin (SM) and saturated glycerophospholipids [21,52,53,54,55], which are organized to form lipid subdomains on EV membranes. Of interest, the composition and distribution of these lipids is expected to be similar to that described for lipid rafts, cholesterol-rich microdomains of cell membranes, which are unique platforms for the segregation and accumulation of specific molecules, including GPI-anchored proteins and other signaling proteins like steroid receptor coactivator (Src) family kinases, to regulate their interactions and activities [56,57].

Emerging data report that the lipid composition of EVs is tightly dependent not only on the different cell types, but also on the activation of specific intracellular pathways, with different regulatory mechanisms, which influence the biogenesis and release of EVs. In this regard, recent studies have suggested that human fibroblasts undergoing senescence show an increased release of EVs, with selective sorting of cellular lipids [34]. Firstly, the vesicular lipid bilayer displays a higher lipid/protein ratio with respect to cells, characterized by a higher level of sphingolipids with respect to glycerophospholipids.

Among the most relevant lipid species (i.e., more than 1% of total detected EV lipids), SM, phosphatidylserine (PS) and lysophosphatidic acid (LPA) have been found to be increased, whereas phosphatidic acid (PA) and phosphatidylethanolamine (PE) have been found to be decreased, suggesting activation of phospholipase A2 or A1 enzymatic activity towards PA in EVs. Studies on reticulocyte maturation suggest that lipid raft domains may participate in molecule segregation during reticulocyte exosome formation. In fact, molecules, including ganglioside GM1, Src tyrosine kinase (Lyn) and proteins containing prohibitin domains (flotillin-1 and stomatin), are sorted as raft components on exosomal membranes [58]. These data agree with previous studies [59], which demonstrated a highly specific enrichment in stomatin, cholesterol and GM1 on MVs shed from erythrocytes after Ca^++^ treatment. However, the raft protein flotillin-1 was not recovered in the analyzed vesicles, suggesting the presence of two distinct types of lipid rafts on the erythrocyte membrane that are specific for stomatin or flotillin segregation, which are involved in the vesiculation process. In addition, sphingosine 1-phosphate (S1P) was observed to regulate cargo (such as CD63, CD81 and flotillin) sorting into exosomes via inhibitory G protein (Gi)-coupled S1P receptors located on MVB membranes [60]. This implicates that exosomal membranes may harbor membrane subdomains enriched in cholesterol and GSLs that contribute in signaling and sorting of specific proteins into EVs/exosomes [61,62].

## 5. Lipids Rafts Involved in EV Biogenesis, Sorting and Secretion

### 5.1. Lipid Rafts

Lipid rafts are defined as highly dynamic microdomains within the lipid bilayer of cellular membranes, which organize signaling molecules into multi-functional complexes or sorting platforms [62]. A key characteristic of raft domains is the compartmentalization of specific components to regulate their interactions with other membrane components, i.e., lipids and proteins, and hence their activity [57,63]. The lipid composition of rafts is distinct from that of the surrounding membrane, with enrichment of cholesterol and sphingolipids, including sialic-acid containing glycosphingolipids (GSLs), which have been suggested as core components of lipid rafts, and are therefore used as typical lipid raft markers [64]. Several works have elucidated that lipid rafts, more specifically known as lipid “raft-like microdomains”, are also located on the membranes of subcellular organelles, including the ER, the Golgi apparatus, endosomes, lysosomes and lipid droplets [65], but also the mitochondria [66,67] and nucleus [68]. At these levels, “raft-like microdomains” contribute to catalyzing key reactions in the context of diverse biological processes, including the regulation of intracellular trafficking and sorting [69], cholesterol homeostasis [70] and cell fate, i.e., survival or death [71,72,73].

Recently, several lines of evidence have supported the role of lipid raft-associated molecules in regulating EV biogenesis and release (Figure 2). Attention has been focused on the lipid composition of EV membranes, which could play a role in conferring stability to these vesicles in different extracellular environments and/or to facilitate binding or uptake into recipient cells.

### 5.2. Lipid Rafts and EVs

Starting from the evidence that the bilayer membrane of EVs contains similar organized lipid microdomains, it is possible to hypothesize that lipid rafts may affect the sorting, secretion, structure and signaling of EVs, and that their heterogeneity can be the result of activation of different biogenesis pathways.

In addition to the pathways’ dependency on the endosomal sorting complexes required for transport (ESCRT) machinery, lipids control many aspects of endosome biology. In fact, MVB biogenesis and EV release can also be achieved by an ESCRT-independent pathway, which is mediated through lipid rafts [74]. In this regard, as mentioned above, a very high level of cholesterol has been found in several exosome preparations. The highly differential distribution of cholesterol in the endocytic pathway is probably the result of efficient lipid sorting processes in the membranes of early- and late-endosomal compartments [75]. Recycling compartments and the internal vesicles of MVBs harbor most of the cholesterol found in the endocytic pathway [76]. Palmulli et al. [77] proposed that in HeLa and MNT-1 melanoma cells, the tetraspanin protein CD63 generates microdomains enriched in cholesterol at endosome-delimiting membranes, which undergo inward budding to generate cholesterol-enriched ILVs, providing a temporary storage site for cholesterol that can be retrieved by the pathway of Niemann–Pick type C1 protein (NPC1). Thus, CD63 emerges as a key regulator of cholesterol endosomal sorting, providing alternative sources of cholesterol through ILVs and EVs. An interesting aspect of how tetraspanins can also influence the cholesterol-enriched microdomain organization is that their interactions with lipid macrodomains involve, in part, the addition to proteins of palmitate, a 16-carbon saturated fatty acid, a key condition that allows for the targeting of tetraspanins to lipid domains [78]; these interactions were found to be disrupted following cholesterol depletion by methyl-β-cyclodextrin (MβCD) treatment.

An additional pathway in intra-endosomal membrane transport and exosome formation was reported by Trajkovic et al. [79], who provided evidence for an alternative pathway for the lateral segregation of cargo within the endosomal membrane and EVs, which was not dependent from ESCRT machinery, but rather from raft-based microdomains. These lipid domains may contain high concentrations of sphingolipids from which ceramides are formed. In particular, activation of a neutral sphingomyelinase-2 removes the head group of sphingomyelin inside the lipid raft to form ceramide that, due to its peculiar cone-shaped structure, may induce spontaneous negative membrane curvature, leading to invagination into the endosome membrane, facilitating membrane blebbing and EV shedding budding, as reported by Menck et al. [80]. Moreover, ceramide seems to be crucial for the generation of a different population of ILVs that are not destined for transport to the lysosomes, but are secreted as exosomes. Additionally, Fukushima et al. [81] proposed a model in which ceramide transfer protein (CERT) facilitates the release of EVs enriched in ceramide under lipotoxic conditions, contributing to the modulation of sphingolipid metabolism [82]. More recently, other authors have emphasized the involvement of CERT in the biogenesis of a unique subset of RNA-containing EVs [83]. By exploring the role of CERT in modulating the sphingolipid composition and biogenesis of EVs under physiological conditions [84], they suggested two distinct mechanisms by which CERT regulates EVs in neuronal cells: (i) the indirect pathway via the trans-Golgi network, which is regulated by phosphatidylinositol 4-phosphate (PI4P) production, and involves SM production through neutral sphingomyelinase-1 (SMase-1) and SMase-2 activity; and (ii) the direct pathway via ER-endosome contact sites, regulated by the interaction of CERT with Tsg101 protein, which is part of the ESCRT-I complex. They present, for the first time, evidence that CERT connects ceramide to the ESCRT-dependent pathway for EV biogenesis. Moreover, overexpression of CERT increases EV secretion, whereas its inhibition reduces EV formation and the concentration of ceramides and SMs in EVs.

Another cone-shaped lipid, which is involved in the formation of internal vesicles in liposomes, is LBPA. Although LBPA was undetectable in human B cell-derived exosomes [85], a role for this lipid in regulating biogenesis and dynamics of ILVs along the degradative pathway has been suggested [86].

While lipid rafts have been considered to play a crucial role in EV biogenesis, the connection between protein–raft association and protein loading into EVs has been only recently elucidated. A recent work identified the main characteristics which may facilitate protein cargo loading into EVs, by using spectroscopy and bioinformatic analysis [87]. In this regard, they investigated the link between lipid–protein interaction, trafficking and loading into EVs, supporting a crucial role for lipid rafts. Encapsulation of Src family kinases into EVs may be regulated by different biogenesis pathways. Myristoylation is required for the association of Src kinases with EVs [88,89], while both myristoylation and palmitoylation may contribute to Fyn encapsulation into EVs [90]. This agrees with previous data obtained from a proteomics study, which showed that enrichment of palmitoylated proteins into EVs was suppressed by palmitoylation inhibitors [91].

## 6. EVs and Secretory Autophagy

### 6.1. Role of Autophagy in EV Release

Autophagy is a fundamental intracellular degradation pathway, responsible for maintaining cellular homeostasis by degrading and recycling misfolded proteins, damaged organelles, and other cellular debris. This process can be rapidly activated in response to various physiological and pathological stimuli, including nutrient deprivation, oxidative stress, hypoxia and hormonal regulation [92]. Under normal conditions, autophagy functions as an adaptive mechanism that promotes cell survival by mitigating cellular stress and sustaining energy balance. However, under certain pathological contexts, excessive or dysregulated autophagy can trigger cell death, exhibiting distinct cytological features that may contribute to disease progression [93]. Under nutrient-rich conditions, mammalian target of rapamycin complex 1 (mTORC1) phosphorylates and inactivates the ULK1 kinase, thereby suppressing autophagy. Conversely, during nutrient deprivation, mTORC1 activity is downregulated, leading to autophagy induction [94]. This process is triggered by a protein complex comprising ATG1 and ULK1, which facilitates the recruitment of the class III phosphatidylinositol 3-kinase (PI3K) complex. This macromolecular assembly includes BECLIN1 ATG14L, VPS34 and VPS15, which catalyze the synthesis of phosphatidylinositol 3-phosphate (PI3P), thereby directing autophagic proteins to the site of phagophore formation and expansion [95]. During autophagosome biogenesis, ATG8 (LC3 in mammals) is initially processed by ATG4, and subsequently activated by ATG7. The C-terminal glycine of ATG8 is then conjugated to PE, thus facilitating membrane elongation and eventual closure [96]. Mature autophagosomes can fuse with lysosomes, allowing the degradation of their cargo by acidic hydrolases. Additionally, autophagosomes may interact with endosome-derived structures, such as the MVBs, forming hybrid vesicles known as amphisomes that subsequently merge with lysosomes to generate autolysosomes, further promoting intracellular degradation and recycling [97,98]. This crosstalk between different vesicular trafficking pathways underscores the intricate coordination of cellular processes, particularly during autophagy. Of interest, vesicles from endocytosis and endosomal compartments either direct damaged molecules to lysosomes or shed them as exosomes via exocytosis; in fact, amphisomes can fuse either with lysosomes for cargo degradation or with the plasma membrane to release MVBs, containing ILVs, into the extracellular space as exosomes [99,100].

With regard to this novel aspect of autophagy in vesicular trafficking, MVBs play a pivotal role in cargo processing and fate determination, directing materials toward degradation, recycling or secretion [101,102]. Therefore, beyond its traditional role in degradation, autophagy can also be implicated in a process simply referred to as the “unconventional secretion pathway” or “secretory autophagy”, as described for the first time in the transport of cytokines, such as interleukin 1β (IL-1β), from the cytosol to the extracellular matrix [103]. This form of unconventional secretion can be triggered and increased upon ER stress, starvation, lysosomal dysfunction [104], unfolded protein response [105] and defects in the intracellular trafficking machinery [106], or in response to impaired autophagosome maturation by autophagy inhibitors, such as chloroquine or bafilomycin A1 [107]. New findings indicate that ATG proteins mediate a novel form of defense in response to infection, whereby exosomes are produced which serve as decoys for bacterially produced toxins [108]. Moreover, secretory autophagy regulates the release of proteins that lack peptide signals, but it can also be considered as an alternative pathway to facilitate the transport of proteins toward the plasma membrane. In this way, the secretory pathway cooperates to regulate intracellular balance, intercellular communication and the tissue microenvironment.

Emerging data support the fact that the generation of MVBs represents an important step in cargo processing during autophagy. Evidence suggests that the fate of MVBs, i.e., degradation or secretion, depends on cellular conditions [109]. Murrow [110] demonstrated that suppressing the ATG12–ATG3 complex, which mediates LC3β conjugation, can alter the morphology of MVBs, disrupt late-endosome trafficking and consequently reduce exosome production. Furthermore, inhibition of ALIX [111] also reduces autophagic flux, supporting a regulatory interplay between exosome biogenesis and autophagy pathways. Similarly, autophagosomes, which carry biological cargo, may also be influenced by cellular states. For example, in lung epithelial cells, IFN-γ stimulation can trigger the secretion of Annexin A2 (ANXA2) via the autophagy pathway [112]. This highlights the ability of autophagosomes to switch from degradation to secretion, resembling exosome-based secretion. Colocalization of exosome markers, such as CD63 and CD81, with autophagosomal proteins, such as p62 and LC3, further supports the shared molecular pathways between autophagy and exosome biogenesis in the generation of amphisomes [113,114].

The biogenesis of amphisomes has gained growing interest, and multiple key molecules, including SNARE proteins, Rab-GTPases and ESCRT, have been indicated as important regulators that drive this process [115].

Rab family small GTPases are essential regulators of intracellular vesicle trafficking, including the transport of exosomes, multivesicular bodies (MVBs), amphisomes and lysosomes [116]. Rab11 was the first Rab GTPase identified to mediate exosome secretion by facilitating the recruitment of MVBs to form amphisomes and subsequently promoting their fusion with the plasma membrane, as observed in K562 cells [100].

Emerging evidence indicates that secretion of IL-1β occurs through a Rab8a-dependent mechanism, highlighting the role of Rab proteins in the regulation of secretory autophagy machinery [103]. The GTP-bound active form of Rab37 is a further key molecule in the vesicle trafficking directionality, supporting a functional role in Rab-mediated vesicle dynamics. Notably, Rab37 activation has been reported to direct vesicle trafficking toward the extracellular secretion of tissue inhibitor of metalloproteinase 1 (TIMP1) in lung cancer cells under starvation, as well as for insulin secretion, likely through increased MAP1LC3/LC3 [116]. In parallel, Rab8a and Rab27a regulate the transport and fusion of amphisomes with the plasma membrane to facilitate the release of high-mobility group box 1 protein (HMGB1) during autophagy. Additionally, Rab35 facilitates exosome release via a synaptosome-associated protein-dependent mechanism [117].

Once activated and bound to membranes, Rab proteins interact with various effectors, including SNAREs, which are essential mediators of membrane fusion events. SNARE proteins form complexes that coordinate the merging of distinct membrane compartments—a fundamental process in autophagy. Specific SNARE complexes, such as STX17–SNAP29–VAMP8 and YKT6–SNAP29–STX7, facilitate the fusion between autophagosomes and lysosomes; the autophagic protein ATG14 is also involved in recruiting and assembling these SNARE complexes at autophagosomal membranes during autophagosome maturation [118].

### 6.2. Role of Lipid Rafts in EV Release During Secretory Autophagy

Bioactive sphingolipids regulate autophagy by exerting their effect in different steps of the autophagy pathway. As mentioned above, the ESCRT-independent machinery, which includes different molecular lipids, tetraspanins and microdomains situated on distinct intracellular membrane compartments, are involved in MVB membrane inward budding and exosome sorting [79,119,120] during the secretory autophagy process. Recent data support enrichment of raft components within EV cargos, released from human fibrosarcoma cells (2FTGH) into the extracellular milieu following autophagy triggering [47]. Based on previous data showing that MAMs (lipid raft subdomains on ER membranes in close contact with mitochondria) are pivotal in early autophagosome formation, disialoganglioside GD3 (a raft marker) and ERLIN1 (an ER marker) interaction was found to regulate autophagosome biogenesis. Moreover, GD3 and ERLIN1 co-fractionated with LC3-II in exosome iodixanol gradient fractions [47]. In addition, electron microscopy analysis revealed an association between GD3 and LC3-II. This suggests that autophagy plays a role in enriching raft components within secreted exosomes, indicating a functional link between the autophagic processes and lipid raft-like composition of exosome membranes. These findings confirm that autophagosomes originate from omegasome-like structures, potentially from the ER, and fuse with the plasma membrane to release their contents. Furthermore, the membranes of exosomes display a lipid composition similar to that of lipid raft-like microdomains [58], suggesting that exosomes may be directly formed from lipid rafts.

Ceramide is an additional lipid molecule which characterizes EVs; it is a precursor to more complex sphingolipids, such as glucosylceramide, ceramide-1-phosphate (C1P), sphingomyelin and galactosyl ceramide. Moreover, following the hydrolysis of SM, ceramide can be phosphorylated by ceramide kinase to form C1P [121], leading to Ca^2+^-dependent liposomal fusion, which increases vesicle fusion in the autophagy pathway. Moreover, depletion of SMase-2 impairs the incorporation of LC3 into endosomes, thereby impeding LC3-dependent EV loading and secretion [114].

In agreement with the role of phosphoinositides (PIPs) in membrane dynamics, in controlling autophagy and in vesicular transport [122], it is not surprising that these lipids may also affect exosome secretion. In particular, phosphatidylinositol-3,5-bisphosphate (PI(3,5)P2), a PIP produced from PI(3)P by the action of the PIP kinases and associated with MVBs [123], is a lysosome-localized phosphoinositide that regulates autophagy during nutrient deprivation [124,125]. It has been found that FYVE-type zinc finger-containing phosphoinositide kinase (PIKfyve), which provides all of PI(3,5)P2 [126], can divert proteins from the pathway of autophagic degradation to that of exosomal secretion in prostate cancer cells [127]. Inhibition or downregulation of PIKfyve can prevent autophagic flux by reducing the levels of PI(3,5)P2 and increase exosome secretion in PC¬3 cells, through a secretory autophagy pathway. These data are in agreement with the observation that PI(3,5)P2 is the only known endogenous agonist which binds and activates the primary lysosomal Ca^2+^ channel, i.e., the transient receptor potential mucolipin 1 (TRPML1) [128], implicated in phagosome–lysosome fusion and autophagosome–lysosome fusion [129]. It has also been further established that inhibition of the class I PI(3)P kinase or Akt, a downstream component in PI3K signaling, increased the exosome release of angiopoietin-2 from lung endothelial cells [130]. A local increase in PI(3)P controls the membrane recruitment of a variety of proteins with PI-binding domains associated with autophagy initiation, such as WIPI proteins (WD-repeat protein interacting with PI), among which WIPI2 has been characterized as an effector of autophagy [131]. Furthermore, it has recently been reported that the inhibition of Vps34 (a class III PI3K) leads to profound dysregulation of lipid metabolism, affecting PI(3)P production [132]. As a consequence, the reduction in PI(3)P levels inhibits autophagy initiation, impairs lysosomal degradation and promotes the secretion of exosomes enriched in undigested lysosomal products, including amyloid precursor protein C-terminal fragments (APP-CTFs), specific sphingolipids and the phospholipid bis(mono-acylglycero)phosphate (PBP), which normally resides in the internal vesicles of endolysosomes. Secretion of these exosomes requires neutral SMase-2 and sphingolipid synthesis. A role for PI(3)P association with ganglioside GD3 and LC3 was also demonstrated in primary fibroblasts upon autophagy triggering [133], suggesting a role for GD3 in autophagosome biogenesis.

## 7. EVs and Apoptosis

### 7.1. Role of Apoptosis in EV Release

Apoptosis is a highly regulated and well-established cellular process that drives cell fate by coordinating a cascade of molecular events to eliminate damaged or unnecessary cells in a controlled manner. This process is characterized by distinct morphological features, including cell shrinkage, nuclear condensation, organelle fragmentation and membrane blebbing. Apoptosis is mediated by caspase enzyme activation via three pathways: the extrinsic pathway, triggered by death ligand–receptor binding; the intrinsic (mitochondria-dependent) pathway, regulated by BCL-2 family proteins; and the ER stress-related pathway, initiated by calcium imbalance and the accumulation of misfolded proteins. The extrinsic pathway activates caspase cascades through death receptor engagement, while the intrinsic pathway promotes mitochondrial outer-membrane permeabilization and subsequent caspase activation.

Finally, the ER stress-induced apoptotic pathway involves cellular responses to proteostasis disruption, further expanding the regulatory complexity of apoptosis. Beyond traditional ApoBDs, apoptotic cells secrete particular vesicles collectively referred to as apoptotic cell-derived extracellular vesicles (ApoEVs), which vary in size and exhibit selective cargo profiles. Notably, dying cells, including apoptotic ones, release significantly higher quantities of EVs compared to healthy cells. ApoEVs are generated by many types of cells, such as stem cells, immunocytes, precursor cells, osteoblasts and endothelial cells [134]. Interestingly, ApoEVs are thought to share similarities with EVs released from healthy cells, particularly in terms of cargo delivery, which includes apoptotic products involved in cell clearance, as well as immune regulation related to inflammation, autoimmunity and cancer, depending on their specific molecular composition. Though recent research has demonstrated that, in addition to ApoEVs, apoptotic cells can release MVs and exosomes, whose production is not specifically dependent upon the apoptosis program, it is now assumed that membrane blebbing is a prerequisite for apoptotic body formation [135]. Activated caspase-3 not only regulates apoptosis, but also promotes the release of extracellular apoptotic exosome-like vesicles (ApoExos), which influence recipient cells [136]. These ApoExos can modulate mitochondrial membrane proteins through miRNA release, affecting the intrinsic pathway activity and membrane permeability. Additionally, they may directly transport key apoptogenic molecules, such as cytochrome c (cyt c), directly impacting apoptotic regulation in target cells. Recently, a key role for caspase-3 has also been described in mediating fusion events between autolysosomes and the cell membrane, therefore controlling ApoExos secretion, particularly in serum-starved endothelial cells [137,138]. In this context, large autolysosomes derived from the fusion of lysosomes, multivesicular bodies and autophagosomes represent a critical site of ApoExo biogenesis. Additionally, ER stress has been shown to enhance exosome release, which, in turn, regulates the ER stress response by modulating key molecules such as PERK and ATF6.

ApoEVs encapsulate a variety of cellular constituents, including nuclear, mitochondrial and plasma membrane proteins, as well as lipids and nucleic acids, like mRNA, long non-coding RNA, rRNA and miRNA fragments. Given their different cargo, ApoEVs can be characterized as metabolically active structures that provide apoptotic cells with the ability to transduce signals over relatively long distances. In fact, Caruso and collaborators [139], exploring the release of vesicles during apoptosis, suggested that these vesicles are not merely cellular debris, but functionally competent structures that modulate immune responses and promote tissue repair processes. While apoptosis has long been viewed as a form of “silent” cell death, recently, this perspective has evolved. Apoptosis is now recognized to play an active role in signaling to nearby cells, influencing their survival, apoptosis and the remodeling of surrounding tissues.

### 7.2. Role of Lipid Rafts in EV Release During Apoptosis

Lipid rafts are pivotal signaling hubs in apoptosis. Death receptors (e.g., Fas, TRAIL-R) translocate into rafts upon ligand binding, where they trimerize and assemble the death-inducing signaling complex (DISC). Rafts cluster adapter proteins such as Fas-associated death domain (FADD) and procaspases, amplifying extrinsic apoptotic signals [140]. Disruption of raft integrity (cholesterol depletion) can inhibit survival pathways (e.g., PI3K/Akt) and sensitize cells to apoptosis [141]. Therefore, EVs that alter raft composition can influence apoptosis. For instance, delivering ceramide-rich exosomes can promote recipient cell apoptosis by enhancing raft-mediated death signaling.

Apoptosis has been observed to induce significant alterations in the remodeling of membrane lipids, not only at the plasma membrane, but also within intracellular membranes [142]. In particular, elevated levels of mitochondrial ceramide contribute to the loss of mitochondrial outer-membrane integrity [143]. Cardiolipin (CL), a phospholipid primarily localized to the inner mitochondrial membrane, appears to translocate into “raft-like” microdomains at contact sites between inner and outer mitochondrial membranes during apoptosis. This redistribution facilitates the local oligomerization of proapoptotic proteins, including Bid. Next, early apoptotic signaling is associated with alteration of membrane trafficking, leading to the redistribution of CL through its association with the cytoskeleton protein vimentin. Indeed, vimentin and CL have been observed to colocalize on the cell surface of apoptotic cells, generating an immunogenic complex [144,145]. Recently, it has also shown that EVs could deliver beneficial molecules and be crucial for cell communication. In particular, PS exposure is a hallmark of ApoEV surfaces [146,147], serving as a signal that promotes their clearance by phagocytosis. Additionally, surface molecule oxidation enhances the binding of thrombospondin (Tsb) and complement protein C3b, further facilitating the recognition of apoptotic cells by recipient cells. These surface markers are considered key indicators for the detection and characterization of ApoEVs.

## 8. EVs in the Crosstalk Between Autophagy and Apoptosis

EVs, including exosomes, microvesicles and ApoEVs, may play a crucial role in modulating signaling pathways related to both autophagy and apoptosis. For instance, activation of the autophagy process leads human endothelial cells undergoing apoptosis to release immunogenic EVs that are enriched with mitochondrial and ER contents, and are therefore different from classical apoptotic bodies; additionally, their biogenesis is dependent on caspase activation, but independent of Rho-associated, coiled-coil-containing protein kinase 1 (ROCK1) activation [148]. Notably, the loading of specific proteins (e.g., RNA-binding proteins) into ApoEVs often depends on the autophagic machinery [114,138]. Autophagic components, including microtubule-associated protein 3 (LC3), ATG16L1 and LAMP2, can also be released from apoptotic cells in which autophagosomes accumulate. This process depends on caspase-3 activation, indicating that the initiation of apoptosis promotes the release of these autophagic components, supporting crosstalk between the autophagic and apoptotic programs [149]. EVs can influence crosstalk between autophagy and apoptosis by delivering specific cargo that modulates signaling pathways involved in both processes. EVs can carry bioactive molecules (e.g., miRNAs, proteins, lipids) that affect the expression of key regulators like Bcl-2, Bax and caspases, thereby overturning the balance between cell survival and death. Beclin1, identified as a Bcl-2-interacting protein, links autophagy and apoptosis through its BH3 domain. Disruption of the Beclin1–Bcl-2 complex, particularly under stress conditions, such as nutrient deprivation, is mediated by JNK1-dependent phosphorylation of Bcl-2, leading to dissociation from both Beclin1 and BAX. This event triggers caspase-3 activation and the transition from autophagy to apoptosis [150]. Recently it has been reported that small EVs derived from neural stem cells (NSC-sEVs) reduce apoptosis and neuroinflammation in spinal cord injuries by promoting autophagy [151]. In fact, NSC-sEVs upregulate autophagy markers LC3II and Beclin-1, enhance autophagosome formation, reduce damage effects and modulate apoptosis- and inflammation-related proteins in target cells.

The interplay between autophagy and apoptosis is regulated by key proteins, such as caspases, FADD (Fas-associated death domain) and ATG5 (autophagy-related 5). These molecules serve as pivotal nodes in determining cell fate, either promoting survival through autophagy or initiating programmed cell death via apoptosis. Caspases, particularly caspase-8, are central executors of apoptosis and autophagy, and can modulate caspase activity. For instance, autophagosomes can sequester active caspase-8, leading to its degradation in lysosomes, thereby attenuating apoptosis and promoting cell survival [152]. Conversely, activated caspases can cleave essential autophagy proteins such as Beclin-1, ATG5 and ATG7, inhibiting autophagy and steering the cell towards apoptosis [153]. FADD serves as an adaptor protein that recruits caspase-8 to DISCs, initiating extrinsic apoptosis. Beyond this role, FADD and caspase-8 are integral in modulating autophagy. In T cells, the FADD–caspase-8 complex associates with the ATG5–ATG12 conjugate, forming a platform that limits excessive autophagy and prevents necroptosis [154]. Thus, the interplay between caspases, FADD and ATG5 constitutes a regulatory network that determines cell survival or death. While their roles in these pathways are well established, emerging evidence suggests that they may also influence EV dynamics. Furthermore, beyond its canonical role, ATG5 has been demonstrated to be implicated in modulating exosome production [155]. In particular, it can disrupt the acidification of late endosomes by disassociating the V1V0-ATPase complex, leading to increased exosome release. This function of ATG5 promotes migration and metastasis in some cancer models, highlighting its role in EV-mediated intercellular communication.

## 9. Interconnected Pathways Between Autophagy and Apoptosis: A Role for Lipid Rafts

Lipid rafts contribute to the biogenesis of EVs by serving as platforms for promoting the cargo sorting and loading of specific proteins and lipids and the molecular machinery required for vesicle release in the extracellular space. This interplay ensures the selective packaging of signaling molecules, including those involved in autophagy and apoptosis, thereby influencing recipient cell behavior. Thus, the intersection of lipid rafts and EVs leads to a complex network that is able to regulate both autophagy and apoptosis pathways. In particular, apoptotic vesicles, upon uptake, induced autophagy and angiogenesis in target cells [156], indicating a bidirectional crosstalk. Thus, it is possible to hypothesize that EVs serve as mediators linking autophagy and apoptosis: their lipid raft content can drive death or survival signals [157]. Indeed, accumulating evidence indicates that the interplay between autophagy and apoptosis is mediated by the sharing of many key regulators, including different lipid molecules, which represent key components of lipid rafts [158]. Ceramide has been typically associated with growth arrest and apoptosis, whereas S1P is primarily involved in promoting cell proliferation and survival. The antagonistic roles and dynamic balance of intracellular ceramide and S1P have been suggested to determine cell fate [159]. However, the recent literature points out a more complex role of sphingolipids in the interplay between autophagy and apoptosis. Indeed, it has been shown that ceramide-induced autophagy may promote cell death, either through the induction of autophagic cell death [160] or by “switching” off autophagy and inducing apoptosis through the CAPN/calpain-mediated cleavage of ATG5 [161] and/or DISC formation [29]. Interestingly, acid SMase appears to be responsible for the induction of autophagy during amino acid depletion, the prototypical pre-autophagic stimulus [162].

Different mechanisms of apoptosis induction have also been characterized for sphingosine; on the contrary, the induction of autophagy in response to sphingosine has not been fully elucidated. However, it has been suggested that sphingosine accumulation and lysosomal dysfunction may disrupt autophagic flux to mediate the accumulation of autophagosomal membranes for DISC-dependent cell death. S1P was recognized as a promitogenic, lipid-derived second messenger [163]. Subsequent studies clarified that sphingosine kinase 1 (SK1) and sphingosine kinase 2 (SK2) isoforms differentially regulate cell fate. Indeed, while SK1 activity is associated with mitogenic and anti-apoptotic effects, overexpression of SK2 has been demonstrated to promote cell death. SK1 was shown to induce autophagy by two mechanisms: (i) the inhibition of the target of rapamycin signaling independently of the Akt/protein kinase B signaling arm, and (ii) the lack of accumulation of Beclin 1 [164]. Like ceramide, gangliosides, sialic acid-containing GSLs, are involved both in apoptosis and in autophagy. Ganglioside GD3 has been shown to directly permeabilize mitochondria in vitro, as well as trigger mitochondrial permeability, cyt c release and caspase activation. On the other side, ganglioside GD3 contributes to the biogenesis and maturation of autophagic vacuoles, since it interacts with PI3P and can be detected in immature autophagosomes in association with LC3-II, as well as in autolysosomes associated with LAMP-1. Hwang et al. [165] demonstrated that ganglioside treatment induced autophagic cell death of astrocytes by activation of the IKBKB/IKK-NFKB/NF-kB pathway. Furthermore, this signaling has been shown to be dependent on the activation of autophagic flux, as revealed by an increase in both LC3-II and autophagosome formation, as well as in the generation of ROS, inhibition of AKT-MTOR, activation of MAPK1 (mitogen-activated protein kinase1)-MAPK3/ERK2-ERK1 and “lipid raft” integrity [166].

Thus, understanding the role of lipids, in particular, of GSLs and EVs, in regulating crosstalk between apoptosis and autophagy could play a crucial role in cell fate, with significant implications, mainly in the pathogenesis of many diseases (Figure 3).

## 10. Extracellular Vesicles and Diseases: A Focus on Autoimmune Disorders and Cancer

EVs play an essential role in intercellular communication, influencing both physiological and pathological processes. Emerging evidence suggests that EVs contribute significantly to the pathogenesis of autoimmune diseases, as well as cancer or infective diseases. These vesicles act as carriers of bioactive molecules, including proteins, lipids and nucleic acids, facilitating disease progression through complex intercellular signaling networks.

Since many lipid species are fundamental in conferring structural and functional properties to EVs, it is unsurprising that dysregulation of lipid metabolism, related to disease states, alters the composition of EV membranes, influencing their ability to interact with recipient cells. In autoimmune diseases, lipid-rich EVs contribute to inflammation and immune activation. For example, oxidized phospholipids found in EVs from rheumatoid arthritis (RA) patients have been shown to promote synovial inflammation and enhance macrophage activation [167]. Furthermore, EV-associated lipids, such as lysophosphatidylcholine and ceramides, can modulate immune signaling, amplifying the pathological immune responses that are characteristic of autoimmune disorders.

Moreover, in RA, synovial fibroblasts and immune cells release EVs enriched with pro-inflammatory cytokines, such as tumor necrosis factor-alpha (TNF-α) and interleukin-6 (IL-6). These EVs contribute to synovial inflammation, cartilage degradation and immune dysregulation, exacerbating disease progression. Additionally, EVs in RA patients have been shown to carry citrullinated proteins, which may serve as autoantigens, promoting the production of anti-citrullinated protein antibodies (ACPAs), a hallmark of the disease, as well as activating dendritic cells, which, in turn, amplify pro-inflammatory cytokine production [168,169]. The persistence of these inflammatory signals creates a vicious cycle of immune activation, leading to joint destruction and systemic complications.

Similarly, antiphospholipid syndrome (APS), a systemic autoimmune disorder characterized by thrombotic events and pregnancy complications [170], is closely linked to EV-mediated pathogenesis. Studies indicate that endothelial cells and platelets in APS patients release EVs containing procoagulant factors, including tissue factor and annexin V [171]. These EVs enhance thrombus formation by facilitating platelet aggregation and endothelial dysfunction, increasing the risk of vascular complications. Moreover, EVs in APS patients can transfer pathogenic autoantibodies, further amplifying immune activation and coagulation abnormalities. This interplay between EVs and the coagulation cascade highlights their potential role as both biomarkers and therapeutic targets in APS management.

EVs also play a crucial role in infectious diseases, by facilitating pathogen–host interactions, immune modulation and disease progression. Many pathogens, including bacteria, viruses and parasites, exploit EVs to enhance their survival and dissemination. Viral infections, such as HIV and SARS-CoV-2, induce the release of EVs carrying viral components, including proteins and RNA, which contribute to immune evasion and intercellular transmission. Similarly, in bacterial infections, EVs derived from infected host cells can carry bacterial toxins, inflammatory mediators and antimicrobial resistance factors, exacerbating disease severity. Additionally, EVs play a dual role in infection, as they can also modulate host immune responses by delivering antigenic materials that stimulate immune activation. In this context, EVs have been shown to carry Nef protein and viral TAR RNA, which can modulate recipient immune cells by downregulating MHC class I expression and impairing T cell activation through NF-κB pathway suppression. Similarly, EVs from SARS-CoV-2-infected cells have been reported to contain viral spike protein and miR-148a, contributing to the modulation of type I interferon responses and promoting immune evasion. Understanding the role of EVs in infectious diseases could provide novel therapeutic avenues for disease intervention, including EV-based vaccines and targeted drug delivery systems [172].

The role of EVs in cancer is equally of impact. Tumor-derived EVs (TDEs) contribute to cancer progression by promoting angiogenesis, immune evasion and metastasis. These vesicles carry oncogenic proteins, nucleic acids and lipids that modulate the tumor microenvironment. Lipid components of TDEs play a crucial role in tumor progression; for instance, EVs enriched in cholesterol and GSLs have been shown to facilitate cancer cell migration and invasion. Additionally, bioactive lipids, such as prostaglandins within EVs, contribute to immunosuppression, enabling tumor cells to evade immune detection [27]. For instance, TDEs in breast cancer and melanoma have been found to facilitate pre-metastatic niche formation by preparing distant tissues for tumor cell colonization. Additionally, EVs play a role in drug resistance by transferring drug-efflux pumps and resistance-associated microRNAs to neighboring cells, limiting the efficacy of chemotherapeutic agents. This ability of TDEs to reprogram recipient cells extends to immune modulation, where they can suppress anti-tumor immune responses, enabling tumor cells to escape immune surveillance [173].

Moreover, EVs in the tumor microenvironment influence stromal cells and immune cells to support cancer progression. For example, macrophage-derived EVs can promote tumor growth by inducing an M2-like phenotype, which is associated with immunosuppression and tumor promotion. Specifically, these EVs have been shown to carry microRNAs such as miR-21 and miR-223, as well as transforming growth factor-beta (TGF-β), which activate STAT3 and PI3K/Akt signaling pathways in recipient macrophages, driving M2 polarization and enhancing tumor-supportive functions. These complex interactions underscore the multifaceted role of EVs in cancer biology, making them attractive targets for therapeutic intervention [27]. Moreover, EVs released from the tumor microenvironment showed differential and altered proteomic content in response to a variety of stress factors, including reduced oxygen concentration, altered levels of glucose and glutamine, pH variations, oxidative stress and Ca2+ ion concentration [174]. Thus, a systematic analysis of EV cargo released in stressful environments could provide valuable information about the expected responses of recipient cells.

Beyond their pathogenic roles, EVs hold potential as biomarkers and therapeutic agents. The molecular cargo of EVs, including disease-specific microRNAs, proteins and lipids, can serve as diagnostic and prognostic indicators. Growing evidence indicates that EV-associated lipids actively contribute to the functional regulation of target cells. Therefore, exploring the EV lipidome presents a compelling research opportunity, providing deeper insights into the intricate interplay between lipids and other EV components, as well as the involvement of EV-associated lipids in intercellular signaling and (patho)physiological mechanisms. For example, EVs derived from mouse oligodendrocytes, human prostate adenocarcinoma, B cells and hepatocellular carcinoma, among others, are particularly rich in cholesterol. In most of these different EV types, phosphatidylcholine or PC is the second most abundant lipid, except for B cells and human prostate adenocarcinoma-derived EVs, which contain a higher concentration of sphingomyelin or SM than PC [167].

Recent advances in EV research have also led to the exploration of their potential in personalized medicine. By analyzing EV profiles in individual patients, clinicians may be able to tailor treatment strategies based on disease-specific EV signatures. This approach holds promise for autoimmune diseases, where heterogeneity in disease presentation and progression poses challenges for standard treatment protocols. Similarly, in oncology, EV-based liquid biopsies could provide real-time insights into tumor evolution, enabling adaptive treatment strategies to overcome resistance mechanisms.

The lipid components of EVs play a crucial role in their biogenesis, stability and function, making them essential players in disease pathology [175]. Future research should focus on elucidating the specific mechanisms governing EV function in different pathological contexts, paving the way for novel diagnostic and therapeutic strategies. With increasing understanding of EV biology, there is growing optimism that targeting EV pathways could revolutionize the management of autoimmune diseases and cancer, offering more effective and personalized treatment approaches.

## 11. Lipid Raft-Targeting Drugs

EVs display highly organized membranes, which are essential for their biogenesis, stability in the extracellular environment and interaction with recipient cells. Their lipid composition not only contributes to structural integrity, but also serves as a reservoir of bioactive lipid mediators capable of initiating specific signaling pathways. Over the past decade, advancements in EV lipidomics have enabled the discovery of novel lipid species, offering promising opportunities for the identification of new therapeutic targets and the enhancement of drug efficacy.

Cholesterol, the most abundant lipid component of EV membranes [53], plays a pivotal role in exosome formation. Consequently, understanding the mechanisms underlying its transport and secretion has attracted considerable interest [175]. Notably, lipid rafts—membrane microdomains enriched in cholesterol and sphingolipids—are critical for EV production, as their physical and biochemical properties arise from specific molecular interactions [54]. Targeting components of lipid rafts to pharmacologically inhibit EV biogenesis and release is emerging as a compelling therapeutic strategy. Several compounds have been identified that modulate EV formation, trafficking and secretion. Since SM and cholesterol are key determinants of membrane fluidity and the structural stability of lipid rafts, pharmacological agents influencing these lipids have been investigated for their potential to interfere with various steps of EV biogenesis. For instance, SMases, which catalyze the hydrolysis of SM to generate ceramide, are known to promote membrane budding and vesicle formation [176]. GW4869, a widely used inhibitor of neutral SMase, blocks this conversion, thereby inhibiting multivesicular body budding and exosome release. This effect was first demonstrated in live cells through the suppression of microRNA-containing exosomes [177]. Additional studies confirmed that GW4869 can inhibit the in vivo release of pro-inflammatory exosomes from macrophages, neutrophils, platelets and endothelial cells, with beneficial effects on cardiac function [178]. More recently, GW4869 was shown to enhance the fusion of autophagosomes with MVBs to form amphisomes and to block hepatitis B virus (HBV) virion release from hepatocytes, suggesting a potential role of amphisomes in HBV biology [179] and highlighting the involvement of ceramide-mediated EVs in HBV DNA transmission [180]. Conversely, stimulation with exogenous SMases has been proposed as a means to increase exosome production in vitro, particularly in cell lines of interest for research or therapeutic use. Another compound, pantethine, which modulates cholesterol biosynthesis in human fibroblasts [181], was found to inhibit MP shedding from TNF-α-stimulated endothelial cells, thus preventing murine systemic sclerosis [182]. In the context of MVs, imipramine—an antidepressant belonging to the tricyclic class—was shown to block their release from glial cells by inhibiting acid SMase activation, and thereby disrupting ceramide-enriched membrane domains essential for MV generation [183]. Beyond these well-characterized inhibitors, other agents have been explored for their impact on EV dynamics. Statins, best known for their lipid-lowering properties, have also been associated with reduced MV release, likely due to their ability to modulate membrane cholesterol content. In particular, simvastatin has been shown to interfere with exosome biogenesis across various in vitro and in vivo models, suggesting its potential as an adjuvant therapy in exosome-related pathologies [184]. More recently, a novel mechanism involving atorvastatin has been identified: exosomes derived from atorvastatin-pretreated bone marrow mesenchymal stem cells presented enhanced proangiogenic effects on endothelial cells through the AKT/eNOS signaling pathway, mediated by the presence of miR-221-3p on the vesicle surface—highlighting their therapeutic potential in treating diabetic skin wounds.

Overall, combinatorial pharmacological strategies targeting multiple steps in EV biogenesis and release may offer more effective means of intervention in disease settings. Continued research is essential to refine these approaches, enhance specificity and reduce off-target effects, with the ultimate goal of translating EV-targeted therapies into clinical applications.

## 12. Conclusions

Autophagy and apoptosis represent two fundamental pathophysiological mechanisms of cell fate regulation. Nevertheless, the signaling pathways of both these processes are deeply interconnected through various crosstalk mechanisms. This review highlights that these interconnected pathways are crucial for EV shedding, focusing on the role of raft-like microdomains, which represent physical and functional platforms operating during the early steps of both the autophagic and apoptotic process.

By providing an understanding of the molecular machinery that links these processes to vesicular trafficking and lipid metabolism, this review supports the hypothesis that affecting these pathways may contribute to the pathogenesis of a variety of diseases, including autoimmune disorders and cancer. Furthermore, these findings emphasize the complexity of autophagy/apoptosis crosstalk and its key role in cellular dynamics, making it a crucial area for further research and therapeutic exploration.

Inhibition of one pathway may enhance or inhibit another pathway. For instance, treatment of deprived cells with autophagy inhibitors accelerates apoptotic cell death.

However, it could also be of interest to evaluate EV biogenesis and release during the interplay between apoptosis and secretory autophagy, with particular attention paid to the role of lipid raft components in regulating these pathways.

## Figures and Tables

**Figure 1 cells-14-00749-f001:**
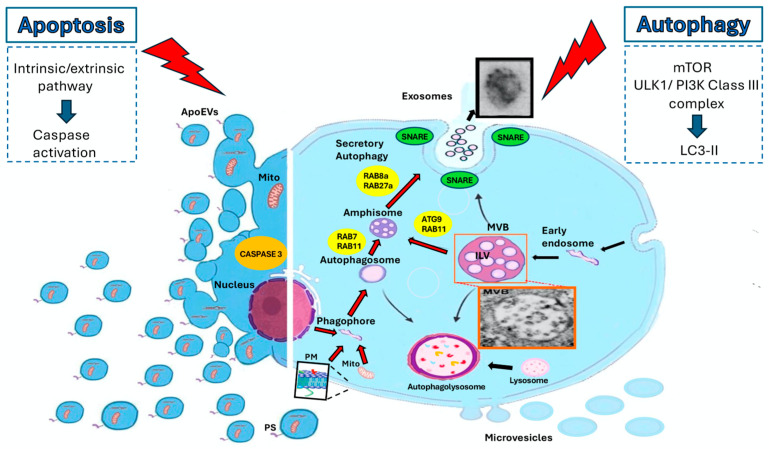
The processes underlying EV biogenesis and secretion have been associated with the activation of distinct signaling pathways, such as those regulating apoptosis and autophagy. Representative transmission electron microscopy (TEM) images show, respectively, an intracellular MVB-like structure containing small internal vesicles, i.e., ILVs (orange box), and extracellular small vesicles, i.e., exosomes (black box), obtained from human fibrosarcoma cells following autophagy triggering, as reported by Manganelli et al. [47]. The principal key pathways in autophagy and apoptosis are shown.

**Figure 2 cells-14-00749-f002:**
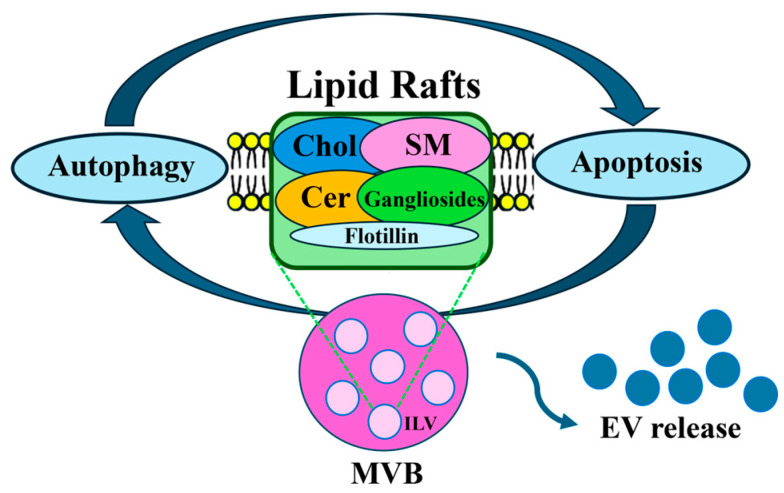
A summary scheme showing the molecular links between lipid rafts, autophagy, apoptosis and EV release.

**Figure 3 cells-14-00749-f003:**
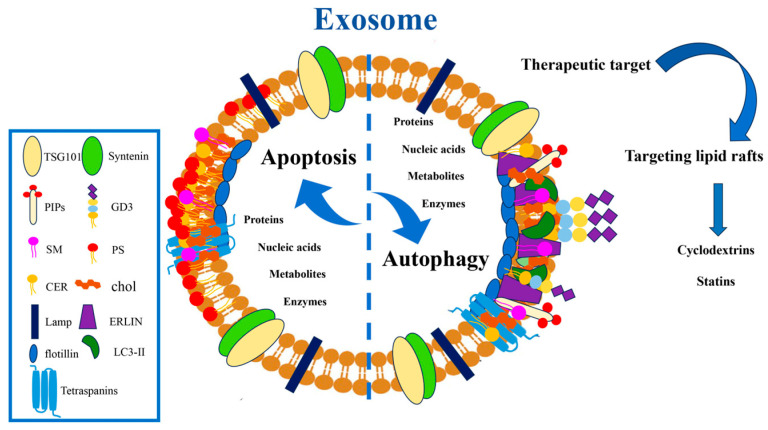
A schematic representation of an exosome enriched with classical exosome markers and lipid raft components involved in the crosstalk between apoptosis and autophagy. Lipid rafts may represent a target for drugs, such as cyclodextrins and statins. Abbreviations: Phosphoinositides (PIPs), Disialoganglioside (GD3), sphingomyelin (SM), Phosphatidylserine (PS), ceramide (CER), cholesterol (chol), lysosomal membrane protein (LAMP), ER lipid raft protein (ERLIN), microtubule-associated protein 1B-light chain 3 (LC3-II), Tumor Susceptibility 101 (TSG101).

## Data Availability

Not applicable.

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
