# Peer review of "Extracellular Vesicles in the Crosstalk of Autophagy and Apoptosis: A Role for Lipid Rafts"

_cells, 2025, doi:10.3390/cells14100749_

Round 1
Reviewer 1 Report
Comments and Suggestions for Authors
This review covers an important and timely topic—how extracellular vesicles are influenced by both autophagy and apoptosis, with a focus on lipid rafts. The authors bring together different areas like membrane biology, lipid metabolism, and cell death in a clear and organized way. It will be a helpful resource for researchers studying EVs, especially those interested in how lipids shape vesicle function.
The paper is well-referenced and includes a lot of useful information, but a few sections could be clearer. Adding some visual summaries would also help readers better understand the complex mechanisms described.
Suggestions:
- A summary figure showing the molecular links between lipid rafts, autophagy, apoptosis, and EV release would strengthen the paper and help visualize the connections described. Also, the resolution of the existing images should be improved further.
- The section discussing amphisomes is interesting but underdeveloped. Please consider adding detail about key regulators (e.g., Rab proteins, SNAREs) and mention their role in controlling amphisomes trafficking and so on.
- The disease section is relevant but a bit too general—it would be stronger with more mechanistic detail- for example, in the case of macrophage-derived EVs in promoting M2 polarization and tumor growth would be stronger with a mention of specific EV components or signaling pathways.
With these revisions, the manuscript will be clearer and more impactful. I support publication after minor revisions.
Author Response
Response to Reviewer 1 Comments
This review covers an important and timely topic—how extracellular vesicles are influenced by both autophagy and apoptosis, with a focus on lipid rafts. The authors bring together different areas like membrane biology, lipid metabolism, and cell death in a clear and organized way. It will be a helpful resource for researchers studying EVs, especially those interested in how lipids shape vesicle function.
The paper is well-referenced and includes a lot of useful information, but a few sections could be clearer. Adding some visual summaries would also help readers better understand the complex mechanisms described.
We thank the reviewer for reviewing the manuscript and for the careful comment on our review. We modified the text following all the helpful suggestions. Please find the detailed responses below and the corresponding corrections highlighted in the re-submitted manuscript.
Comment 1: A summary figure showing the molecular links between lipid rafts, autophagy, apoptosis, and EV release would strengthen the paper and help visualize the connections described. Also, the resolution of the existing images should be improved further.
Response 1: We thank the reviewer for the suggestion and provided a schematic drawing as new Figure 2, line 188 pag. 5; moreover, the resolution of the existing images was improved.
Comment 2: The section discussing amphisomes is interesting but underdeveloped. Please consider adding detail about key regulators (e.g., Rab proteins, SNAREs) and mention their role in controlling amphisomes trafficking and so on.
Response 2: We appreciate the comment of the reviewer and updated the manuscript with the requested information on the role of Rab proteins and SNAREs in the section discussing amphisomes trafficking, pag.8 line 329-352. We also added new references (Ref.115, Ref.116, Ref.117).
Comment 3: The disease section is relevant but a bit too general—it would be stronger with more mechanistic detail- for example, in the case of macrophage-derived EVs in promoting M2 polarization and tumor growth would be stronger with a mention of specific EV components or signaling pathways.
Response 3: We thank the reviewer for pointing this out and we improved the disease section by adding more details, where appropriated. Line 625-626, line 647-652, line 672-675.
Reviewer 2 Report
Comments and Suggestions for Authors
See attached

In line 147, delete the word ..Anyway. Line 211, change ....which was not dependent from ESCRT machinery, but rather from raft-based... to ...which was not dependent on ESCRT machinery, but rather on raft-based...
Author Response
Response to Reviewer 2 Comments
Comment 1: This is a comprehensive review tackling a rather difficult and understudied role of lipid rafts in Extracellular vesicle biogenesis and trafficking as well as the role of EVs in the cross-talk of autophagy and apoptosis. The major problem with the review is the coverage of two broad topics which in the end makes the review diffuse and unfocused. It is the opinion of this reviewer that this should be broken into two parts. The first part should only talk of the role of EVs in the crosstalk between autophagy and apoptosis. The second part can then discuss the role that lipid rafts plays in the biogenesis of EVs and how this process is then tied to the cross-talk between autophagy and apoptosis.
Response 1: We thank the reviewer for reviewing the manuscript and for his/her comment on our review. Accordingly, we revised and divided the Paragraph 6. “EVs and secretory autophagy” as well as the Paragraph 7. “EVs and Apoptosis”. In addition, we have also revised and divided the Paragraph “Crosstalk between autophagy and apoptosis” and added new references (see Ref 140-141; Ref 148-157).
Finally, we moved the paragraph “Lipid rafts” as a part of Paragraph 5: Lipids Rafts involved in EVs biogenesis, sorting and secretion”
You can find the corresponding corrections highlighted in the re-submitted manuscript.
Comment 2: Otherwise the potential readers will be overwhelmed. It would also be nice to explore more about the proteins Beclin-1 and Bcl-2/Bcl-xl in the context of EVs since they play a pivotal role in autophagy and apoptosis. Similarly, Caspase activity, FADD and ATG5 all of which play pivotal roles in the cross-talk between autophagy and apoptosis should be discussed.
Response 2: We thank the reviewer for the suggestions. Accordingly, we added more details about the proteins Beclin-1 and Bcl-2/Bcl-xl, as well as about Caspase activity, FADD and ATG5 in the cross-talk between autophagy and apoptosis; we also added new references: Ref. 150-155. Line 510-542.
Comment 3: Make Figure 1 much larger and include all key pathways in autophagy and apoptosis that are mediated by EVs.
Response 3: We replaced the old Figure 1 accordingly. Key pathways in autophagy and apoptosis have been added and are now included.
Comment 4: Try to improve the flow of information. This lack of flow makes it difficult to follow the main points of the review
Response 4: As suggested by reviewer, we improved the flow information by re-organizing the paragraphs in new sub-paragraphs.
Comment 5: There are numerous grammatical errors that need to be corrected such as line 211 should read …which was not dependent on ESCRT machinery and not …which was not dependent from ESCRT machinery.
Response 5: We are sorry for grammatical errors. We have carefully corrected them in the text.
Reviewer 3 Report
Comments and Suggestions for Authors
Longo et al. described about roles of lipides in EV biology in manuscript “ Extracellular vesicles in the crosstalk of Autophagy and Apoptosis: a role for Lipid Rafts“ . The reviewer think generally, the manuscript was well written, but I have several points to correct as below:
- Line 63. ___ the endocytic pathway regulated mainly by ESRCT complex,__
I think exosomes is secreted by ESCRT-dependent and-independent pathway (Dev Cell 2011 Vol. 21 Issue 4 Pages 708-21, J Cell Sci 2013 Vol. 126 Issue Pt 24 Pages 5553-65, MBO Rep 2021 Vol. 22 Issue 5 Pages e51475, etc). Also, it is ESCRT, not ESRCT.
- Line 67. The authors describe ectosomes in line 60-61, however, there is no ectosomes in this paragraph. Better to organize if the authors delete ectosomes in line 60-61, or describe what is ectosomes in line 67.
- Line 86. __,the transferrin receptor for exosomes,__
I believe usually people use CD63 or CD9 for exosome marker, but not transferrin receptor so much.
- Line 166. and phosphatidylserine (PS) were decreased,
I think, PS is enriched in exosomes (Biochimica et Biophysica Acta 1831 (2013) 1302–1309). In fact, the authors described “PS is a hall mark of ApoEVs “ in line 402.
- Line 208, intraendosomal,
Better to add – and make "intra-endosomal", as "intraendosomal" is not common term.
- Line 322, 2FTGH cells
Better to add explanation what kind of cell 2FTGH is. In line 327, fibrosarcoma cells is described. Is 2FTGH fibrosarcoma cells?
- Line324, GD3 (a raft marker)
Better to spell out GD3 in the beginning, and describe what kind of lipid it is.
- Line 450, 459
Please explain what is DISC.
- Line 460-461.
S1P was already described in line 177, don’t need full name shingosine-1-phosphate again. Also, what is SK1 and SK?
- Line 468, cyt c ?
Cytochrome c was already described in line 418, but no abbreviation. If the authors want to use cyt c, cytochrome c (cyt c) should be described in line 418.
- Line 473-477.
The authors should make it clear what is autophagic cell death, whether autophagy per se or inhibition of autophagy induce cell death.
Author Response
Response to Reviewer 3 Comments
Longo et al. described about roles of lipides in EV biology in manuscript “ Extracellular vesicles in the crosstalk of Autophagy and Apoptosis: a role for Lipid Rafts“ . The reviewer think generally, the manuscript was well written, but I have several points to correct as below:
We thank the reviewer for reviewing the manuscript and for the careful comment on our review. We modified the text following all the suggestions as below.
Comment 1: Line 63. ___ the endocytic pathway regulated mainly by ESRCT complex,__
I think exosomes is secreted by ESCRT-dependent and-independent pathway (Dev Cell 2011 Vol. 21 Issue 4 Pages 708-21, J Cell Sci 2013 Vol. 126 Issue Pt 24 Pages 5553-65, MBO Rep 2021 Vol. 22 Issue 5 Pages e51475, etc). Also, it is ESCRT, not ESRCT.
Response 1: We agree with the reviewer and added the ESCRT-independent pathway. We added a new Ref 26; Line 64.
- Ok, we corrected ESCRT, new Line 63.
Comment 2: Line 67. The authors describe ectosomes in line 60-61, however, there is no ectosomes in this paragraph. Better to organize if the authors delete ectosomes in line 60-61, or describe what is ectosomes in line 67.
Response 2: We preferred to delete ectosomes, since our purpose is to provide only a short general section into EV-classification. Line 60.
Comment 3: Line 86. the transferrin receptor for exosomes, I believe usually people use CD63 or CD9 for exosome marker, but not transferrin receptor so much.
Response 3: We thank the reviewer for pointing this out and we modified accordingly. Line 88.
Comment 4: Line 166. and phosphatidylserine (PS) were decreased; I think, PS is enriched in exosomes (Biochimica et Biophysica Acta 1831 (2013) 1302–1309). In fact, the authors described “PS is a hall mark of ApoEVs “ in line 402.
Response 4: We agree with the reviewer and changed in the text as below: “Among the most relevant lipid species (i.e. more than 1% of total EV detected lipids), SM, phosphatidylserine (PS) and lysophosphatidic acid (LPA) were increased”; Line 147.
Comment 5: Line 208, intraendosomal. Better to add – and make "intra-endosomal", as "intraendosomal" is not common term.
Response 5: Done; Line 219.
Comment 6: Line 322, 2FTGH cells; Better to add explanation what kind of cell 2FTGH is. In line 327, fibrosarcoma cells is described. Is 2FTGH fibrosarcoma cells?
Response 6: We now better claryfied that 2FTGH cells are referring to “Fibrosarcoma cells”.
Line 362.
Comment 7: Line324, GD3 (a raft marker) Better to spell out GD3 in the beginning, and describe what kind of lipid it is.
Response 7: We spelled out GD3 in “Disialoganglioside GD3”. Line 365.
Comment 8: Line 450, 459 Please explain what is DISC.
Response 8: Ok, done. Line 468.
Comment 9: Line 460-461. S1P was already described in line 177, don’t need full name shingosine-1-phosphate again. Also, what is SK1 and SK?
Response 9: We replaced the full name shingosine-1-phosphate with S1P, line 572. In addition, we explained SK1 and SK2 accordingly. Line 574.
Comment 10: Line 468, cyt c ? Cytochrome c was already described in line 418, but no abbreviation. If the authors want to use cyt c, cytochrome c (cyt c) should be described in line 418.
Response 10: We now explained cyt c. Line 445.
Comment 11: Line 473-477. The authors should make it clear what is autophagic cell death, whether autophagy per se or inhibition of autophagy induce cell death.
Response 11: We thank the reviewer for pointing this out and clarified with the following sentence: “this signaling has been shown to be dependent on the activation of autophagic flux, as revealed by the increase of both LC3-II and autophagosome formation. Line 587-588.
Round 2
Reviewer 2 Report
Comments and Suggestions for Authors
The authors addressed all my concerns adequately. In my opinion this review will enhance our knowledge and lead to new research directions in EV research.
Comments on the Quality of English LanguageAcceptable